# Intrinsic anharmonic localization in thermoelectric PbSe

M.E. Manley [1], O. Hellman[2], N. Shulumba[2], A.F. May [1], P.J. Stonaha[1], J.W. Lynn [3], V.O. Garlea[4], A. Alatas [5], R.P. Hermann [1], J.D. Budai [1], H. Wang[1], B.C. Sales[1] & A.J. Minnich[2]

Lead chalcogenides have exceptional thermoelectric properties and intriguing anharmonic lattice dynamics underlying their low thermal conductivities. An ideal material for thermoelectric efficiency is the phonon glass–electron crystal, which drives research on strategies to scatter or localize phonons while minimally disrupting electronic-transport. Anharmonicity can potentially do both, even in perfect crystals, and simulations suggest that PbSe is anharmonic enough to support intrinsic localized modes that halt transport. Here, we experimentally observe high-temperature localization in PbSe using neutron scattering but find that localization is not limited to isolated modes – zero group velocity develops for a significant section of the transverse optic phonon on heating above a transition in the anharmonic dynamics. Arrest of the optic phonon propagation coincides with unusual sharpening of the longitudinal acoustic mode due to a loss of phase space for scattering. Our study shows how nonlinear physics beyond conventional anharmonic perturbations can fundamentally alter vibrational transport properties.

[1] Material Science and Technology Division, Oak Ridge National Lab, Oak Ridge, TN 37831, USA. [2] Division of Engineering and Applied Science, California Institute of Technology, Pasadena, CA 91125, USA. [3] NIST Center for Neutron Research, National Institute of Standards and Technology, Gaithersburg, MD 20899, USA. [4] Neutron Scattering Division, Oak Ridge National Lab, Oak Ridge, TN 37831, USA. [5] Advanced Photon Source, Argonne National Laboratory, Argonne, IL 64039, USA. Correspondence and requests for materials should be addressed to M.E.M. (email: manleyme@ornl.gov) or to A.J.M. (email: aminnich@caltech.edu)

Thermoelectric materials have attracted intense interest in recent decades owing to their promise in energy applications, including converting waste heat into electricity and replacing mechanical cooling systems with more environmentally friendly solid-state devices[1]. The challenge has been to improve the low efficiency of the energy conversion process, which is characterized by the dimensionless figure of merit, $zT$, defined as the ratio of the electronic power factor and the thermal conductivity of the thermoelectric material[1]. Lead chalcogenides make good thermoelectric materials[2–7] because they have both high electronic power factors and low thermal conductivities owing to strongly anharmonic lattice dynamics[8–12]. An important strategy for improving the figure of merit is to reduce the phonon contribution to thermal transport while keeping the electrical conductivity unchanged, i.e., using the "phonon glass-electron crystal" approach[13–15]. Approaches have included increasing phonon scattering and/or localizing vibrations by introducing loosely bound rattling atoms in open-structured materials such as clathrates and skutterudites[13,14], nano-precipitates in PbTe[16], nanostructured layered materials[17], engineered disorder[18], and dislocations in PbSe[19]. However, deviations from crystalline order may also scatter or localize electrons, thereby reducing the electrical conductivity component of the power factor. Hence, it is desirable to find phonon blocking behavior, such as strong phonon scattering and localization, in defect-free thermoelectric crystals.

Although anharmonicity is known to reduce thermal conductivity by increasing phonon scattering rates, it is less well known that anharmonicity can also localize vibrational energy even in a perfect crystal[20–24]. The basic concept is an isolated point-defect-like intrinsic localized mode (ILM)—also known as discrete breather—which is a spatially localized vibration that forms due to interplay of discreteness and anharmonicity[20–24]. Anharmonicity causes a change in the local interatomic forces in the vicinity of a local amplitude fluctuation, shifting the frequency into gaps or above the cutoff in the spectrum that exist because of discreteness in the atomic lattice. Once the local vibration is outside the bands and it no longer resonates with the normal phonons, it can persist independently as an ILM. Hence, just like classic impurity modes, ILMs appear as dispersionless modes outside of the phonon bands. Anharmonicity can also result in more complex dynamical patterns that can be thought of either as an interference pattern in extended mode instabilities or as a superlattice of ILMs[25].

An important open question is whether anharmonicity can drive the localization of lattice vibrational energy within the phonon bands in a way that is analogous to impurity resonance modes. An impurity resonance mode occurs when heavy impurities are inserted in a crystal and their vibrations appear dispersionless (localized) over most of reciprocal space, but exhibit anticrossings with the plane wave phonons. Anticrossings alter the phonon velocities rather than simply increasing phonon scattering rates[26].

Recent ab initio molecular dynamics calculations of PbSe that explicitly account for strong anharmonicity produce what appears to be an ILM forming at high temperatures and in resonance with the acoustic modes[10].

Here, we report observations of localization and related changes in the lattice dynamics in a PbSe crystal using inelastic neutron and x-ray scattering. Our results reveal that localization occurs at temperatures close to predicted but involves more spectral weight than expected and drives unanticipated changes in the lattice dynamics including an unexpected sharpening of the longitudinal acoustic (LA) phonon at high temperatures. Rather than localization occurring with a fraction of the intensity of the normal phonons, as predicted[10], the entire spectral weight of a large portion of the transverse optic phonon abruptly develops flat dispersion (zero group velocity) and appears fragmented in frequency. The localization (flattening) and fragmenting of the optic phonon is explained in terms of a transition in the anharmonic dynamics[27,28], which is also detected as a small kink in our thermal diffusivity measurements similar to that observed with ILM ordering in NaI[28]. The observation of in-band localization in a PbSe crystal not only expands the domain of anharmonic localization, but also has important ramifications for the low thermal conductivity[10] that is critical to its thermoelectric efficiency[1]. We find that the rearrangement of spectral features that comes with localization also fundamentally changes the phase space for scattering, which explains the sharpening of the LA phonon. These results show that nonlinear physics beyond conventional anharmonic perturbations can play an important role in controlling vibrational transport.

## Results

**Inelastic neutron scattering.** The neutron scattering measurements shown in Fig. 1 were obtained using the BT7 triple-axis thermal spectrometer at the NIST Center for Neutron Scattering (see Methods), and demonstrate the most prominent changes in the dynamical structure of PbSe on heating from 300 to 793 K. The most dramatic change in the phonon dispersion curves (Fig. 1a) is the complete flattening of the TO phonon dispersion on heating from 643 K (green circles) to 793 K (magenta triangles). This flattening indicates that the TO phonon does not propagate at this high temperature. This is the same temperature range that ab initio simulations predict for an ILM splitting off from the TO phonon carrying about half of its intensity[10]. In contrast, the dispersion curves for the LA and transverse acoustic (TA) phonons show relatively small energy changes with increasing temperature.

Figure 1b–d shows how the TO phonon spectral intensity distribution changes along momentum transfer $\mathbf{Q} = [H, H, 3]$ with temperature. At 300 and 643 K, where the TO mode is dispersive, the intensity is highest near the (113) zone center. The energy-resolution (1.3 meV) corrected linewidths at $H = 0.8$ increase from 2.47 meV at 300 K to 4.3 meV at 643 K, indicating an increase in phonon scattering with increasing temperature. At 793 K, where the dispersion becomes flat, the intensity distribution is also mostly flat, shifting away from the (113) zone center. There is also additional intensity appearing at the bottom of the spectrum near 3–4 meV, which may be a new feature or more likely intensity from the TA phonon spreading up from lower energies. Otherwise, there is no clear evidence of the predicted splitting in the spectrum within the instrument energy resolution of 1.3 meV, although the energy width (~5 meV) is much broader than the resolution and covers that expected for both the TO phonon and the ILM in the simulations[10]. The lower edge (~6 meV) is close in energy to the predicted ILM energy of about 5.5 meV and the upper edge is near the upper edge of the predicted TO phonon[10], suggesting that the difference between simulation and measurement may in part be the sharpness of the split features. To check for any fine energy structure missed with thermal neutrons we obtained high-resolution measurements using the HYSPEC spectrometer at the Spallation Neutron Source (see Methods).

The time-of-flight cold neutron scattering measurements shown in Fig. 2a, b provide a high-resolution (~0.3 meV) view of the dynamical structure of PbSe along $\mathbf{Q} = [H, H, 3]$ near the (113) zone center at temperatures of 294 and 760 K. As expected, the measurements at 760 K (Fig. 2a) yield an overall spectral distribution that appears to be between that observed at 643 and 793 K in the triple axis measurements (Fig. 1b, c). However, the

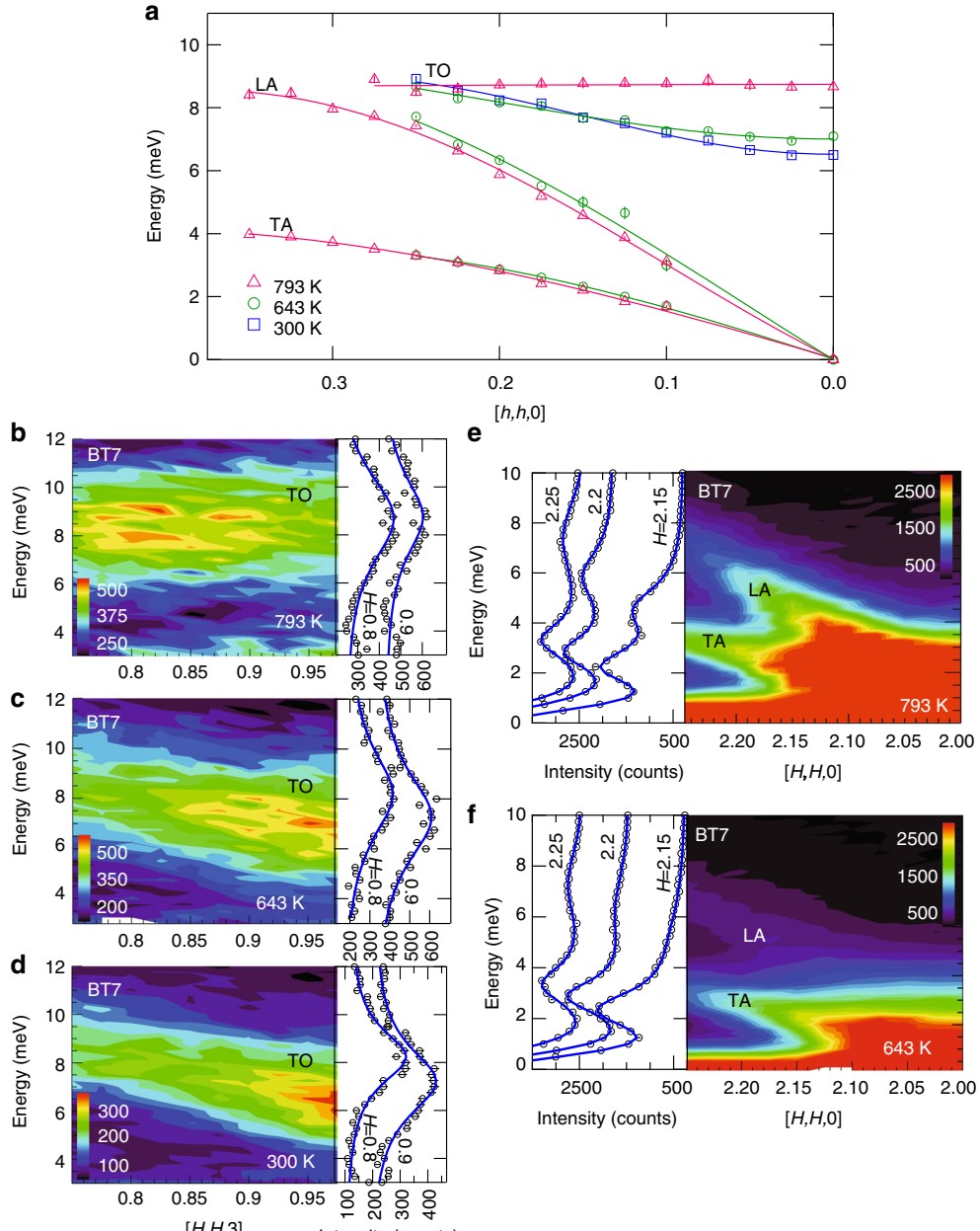

**Fig. 1** Triple-axis thermal neutron scattering measurements of the phonons PbSe. **a** Summary of the temperature dependence of the measured phonon dispersion for the transverse acoustic (TA), longitudinal acoustic (LA), and transverse optic (TO) phonons folded into the first zone along phonon wavevector $\mathbf{q} = [h, h, 0]$. The data points were determined by fitting Lorentzians to the data. The flattening of the TO phonon at 793 K indicates a phonon group velocity, $v_g = dE/dq$, that goes to zero (localization). **b–d** Temperature dependence of the spectral intensity distribution for the TO phonon measured in the (113) zone along scattering wavevector $\mathbf{Q} = [H, H, 3]$. Data sets are offset for clarity. **e, f** Temperature dependence of the spectral intensity distribution for the LA and TA phonons. The TA appears despite the longitudinal geometry because of finite $\mathbf{Q}$ resolution effects. Surprisingly, the LA phonon is sharper at 793 K than at 643 K. Intensity bars are in counts. Error bars are statistical and represent one s.d.

high-resolution measurement reveals an additional sharp but weak dispersionless feature near 5.5 meV (indicated by arrow in Fig. 2a). This feature has an energy width comparable to the instrument resolution of ~0.3 meV. We also confirmed that this feature appears on the neutron energy gain side of the spectrum (see Supplementary Fig. 1). Although it is much weaker, it matches the in-band ILM feature predicted in ref. [10]. Figure 2c shows the result of an ab initio molecular simulation following the procedure in ref. [10] but with additional corrections made for the neutron scattering structure factors (see Methods). The simulation overestimates the intensity of the ILM feature and

underestimates the intensity of the distributed spectral weight associated with the TO phonon that dominates the observed dynamical structure factor. This measured TO spectrum is reminiscent of that in NaI, where a fragmentation and flattening of the spectrum was attributed to conventional gap ILMs ordering in a dynamical pattern[27]. The high-energy resolution measurements of the TO phonon at 294 K (Fig. 2b) are in good agreement with the triple-axis measurements at 300 K (Fig. 1d) and do not show evidence of the large splitting predicted by simulations (Fig. 2d), although there is the hint of a weak feature just below the TO phonon near the (113) zone center that is also evident in

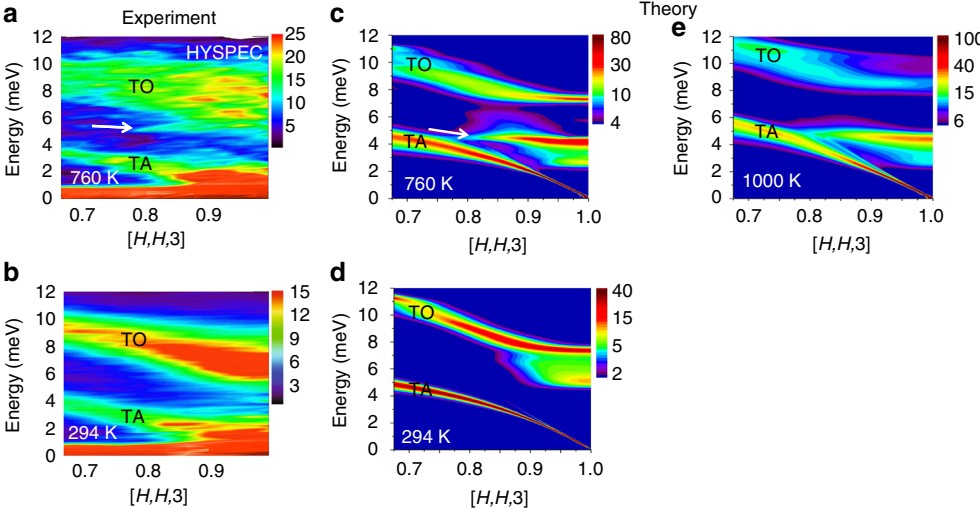

**Fig. 2** Time-of-flight cold neutron scattering measurements and ab initio simulations. These measurements resolve fine energy structure in the spectral distribution for comparison with the ab initio simulations. **a, b** Measurements of the TO and TA phonon spectral intensities along **Q** = [H, H, 3] at 294 and 760 K (out-of-plane direction integrated ±0.1 r.l.u.). **c, d** Ab initio molecular dynamics simulation predictions of the same phonon spectral intensities. Intensity bars are in arbitrary units. **e** Simulation at 1000 K

the simulation. Since uncertainties in the simulation temperature scale exist[10], we also include an additional simulation at 1000 K, Fig. 2e. This spectrum shows a flatter, more fragmented TO phonon, more like the observations. Furthermore, there is a shift of TO phonon spectral intensity away from the (113) zone center, similar to a shift observed in going from 643 to 793 K in the triple axis measurement (Fig. 1b, c). There is also a stiffening of the TA phonon at 1000 K in the simulation, which supports the idea that the additional intensity appearing at the lowest energies in Fig. 1b is from the TA phonon shifting up into the measured range.

As shown in Fig. 1e, f, a large anomalous decrease in the energy linewidths is observed for the LA phonon measured along **Q** = [H, H, 0] with heating from 643 to 793 K, the same temperature where localization of the TO phonon occurs (Fig. 1a–d). Typically, phonon lines broaden with heating as scattering rates increase with increasing phonon population, but in this case the opposite occurs. The LA phonon is broader at 643 K than at 793 K (c.f. Fig. 1e, f). The TA phonon also appears in this spectrum even though it should be excluded by the polarization factor in longitudinal geometry. The TA appears because the relaxed out-of-plane angular (or **Q**) resolution of the BT7 spectrometer introduces a transverse polarization component near the zone center. This "forbidden" TA mode plus the background intensity makes determining energy linewidths challenging, especially at the lower energies. Therefore, to better quantify linewidths we performed additional measurements using inelastic x-ray scattering (IXS), which has the benefit of reduced background and better out-of-plane angular resolution.

**Inelastic x-ray scattering**. The IXS measurements shown in Fig. 3 were obtained on the HERIX-30 instrument at the Advanced Photon Source (see Methods), and provide an accurate measure of the LA phonon energy linewidths at 294 and 770 K. The phonons in the spectra in Fig. 3a were fit using Lorentzians with adjustable widths convoluted with a fixed 1.5 meV width instrument resolution function (see Supplementary Fig. 2). The "forbidden" TA mode is still present but relatively weaker than in the triple-axis neutron measurements (Fig. 1e, f) and the background scattering is negligible. Figure 3b shows the Lorentzian component of the linewidths for the LA phonon at both temperatures.

The LA phonon linewidths are all about a factor of two larger at 294 K than at 770 K and increase slightly closer to the (220) zone center at lower energies. The determined dispersion of the LA mode, shown in Fig. 3c, is in good agreement with the neutron result (reproduced from Fig. 1a). The change in linewidth, $\Gamma$, indicates that the LA phonon lifetime ($\tau = \hbar/\Gamma$) is twice as long at 770 K as it is at 294 K, which has significant implications for the thermal conductivity since this is a high velocity mode. To understand how the LA phonon scatters less frequently at high temperatures despite the increase in phonon population we turn to ab initio molecular dynamic simulations.

**Scattering phase space calculations**. In addition to the phonon thermal population factors, the temperature dependence of phonon scattering rate is governed by a phase space of possible scattering processes that conserve energy and momentum[29]. It is dominated by three-phonon processes, where energy and momentum conservation require frequencies $\omega_1 \pm \omega_2 = \omega_3$ and momenta $\mathbf{q}_1 + \mathbf{q}_2 + \mathbf{q}_3 = \mathbf{G}$, where **G** is a reciprocal lattice vector. The manifold of all allowed scattering channels is what we denote the scattering phase space. When the temperature dependence of the phonon dispersion curves is weak, this manifold can be considered constant and the temperature dependence of the phonon linewidths is controlled by the occupation factor, leading to the expectation that linewidths increase with increasing temperature. In PbSe, however, the scattering phase space changes with temperature, giving rise to the unexpected increase in phonon lifetime with increased temperature (Figs. 1e, f and 3). This change is illustrated in Fig. 4, where by choosing the first frequency to be the TO mode at the zone center, $\omega_1 = TO(\mathbf{\Gamma})$, we consider scattering with the LA mode, $\omega_2 = LA(\mathbf{q}_2)$, and other TO modes, $\omega_3 = TO(-\mathbf{q}_1 -\mathbf{q}_2)$. The phase space for this type of scattering is the surface shown in Fig. 4. As is evident, increasing temperature shrinks the available phase space for scattering, which explains the anomalous sharpening of the LA linewidths.

**Thermal diffusivity and structure analysis**. The abrupt flattening and fragmenting of the TO phonon on heating from 643 to 793 K (c.f. Fig. 1) suggests a transition in the anharmonic dynamics. If ILMs organize in PbSe like they appear to in NaI[27,28]

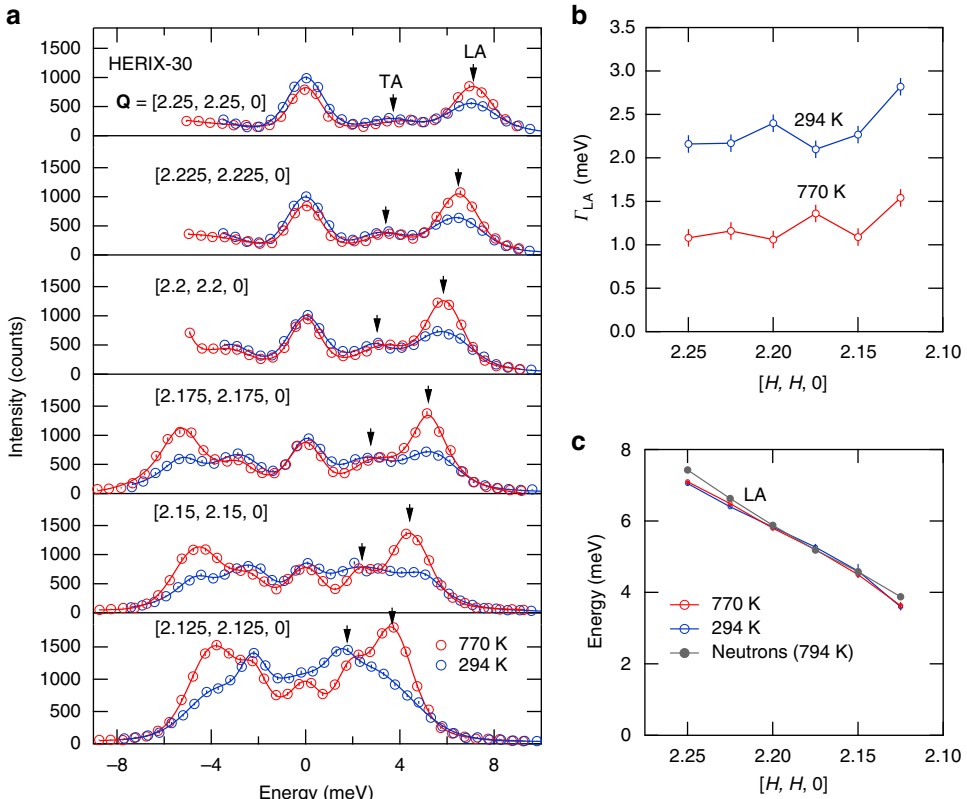

**Fig. 3** Phonon sharpening measured using inelastic x-ray scattering. These measurements provide an essentially background free measure of the anomalous sharpening longitudinal acoustic (LA) phonon at high temperatures in Fig. 1e, f. **a** Spectrum at $T = 294$ and 770 K (circles) along with fits (lines) to the data. Error bars are statistical and represent one s.d. The transverse acoustic (TA) mode also appears but is weaker than with the triple axis measurements (Fig. 1e, f) because of a better out-of-plane **Q** resolution. **b** Resolution corrected energy linewidths for the LA phonon, $\Gamma_{LA}$, full width at half maximum (FWHM). **c** The LA phonon dispersion showing good agreement with the triple-axis neutron results

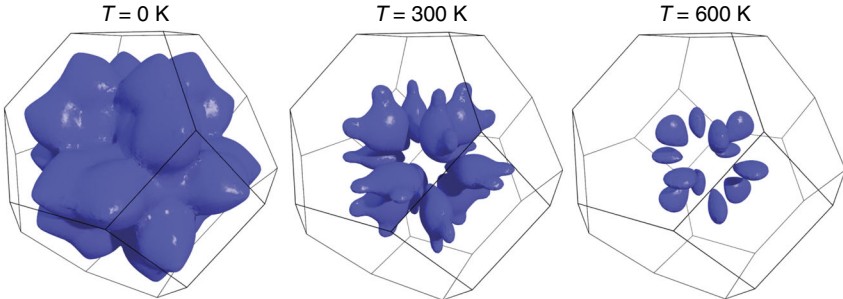

**Fig. 4** Calculated phase space for three-phonon scattering processes in PbSe. This surface in the Brillouin zone represents the three-phonon scattering processes allowed by energy and momentum conservation for the transverse optic (TO) mode fixed at the zone center, $\omega_1 = TO(\mathbf{\Gamma})$, the longitudinal acoustic (LA) phonon, $\omega_2 = LA(\mathbf{q}_2)$, and the TO phonon, $\omega_3 = TO(-\mathbf{q}_1 - \mathbf{q}_2)$. The surface represents allowed values for $\mathbf{q}_2$ for a fixed $\mathbf{q}_1$ ($\mathbf{q}_3 = -\mathbf{q}_1 - \mathbf{q}_2$ is not shown). The resulting reduction in scattering phase space with increasing temperature provides an explanation for the observed decrease in LA phonon scattering rate and linewidth at high temperatures despite the increase in phonon population

then this could explain the flat TO phonon. This is illustrated with a simple model in ref. [27] using an array of soft force constants to represent an ILM superlattice in NaI; the result is a local mode, a flat optic band, and optic mode fragmentation[27]. The physical reason for the flattening is that everywhere the optic mode wavevector matches a multiple of the superstructure of ILMs a standing wave results (indirect localization), and this tends to flatten the optic band overall. In other words, the phonons become trapped between ILMs. Because such dynamical pattern transition enthalpies/entropies are very small they can be missed by conventional scanning calorimetry methods, which

tend to smear out small features, but the laser flash method was shown to be sensitive to these transitions in single crystals of NaI[28]. Therefore, we performed additional thermal diffusivity measurements on our PbSe crystal using a laser flash system (see Methods).

Figure 5a shows that there is a small negative kink in the thermal diffusivity measured on our crystal at 750 K. This feature is at the right temperature to explain the flattening and fragmenting of the TO phonon at 793 K. To rule out any overlooked structural transition across this temperature we also examined the single crystal diffraction pattern from the HYSPEC

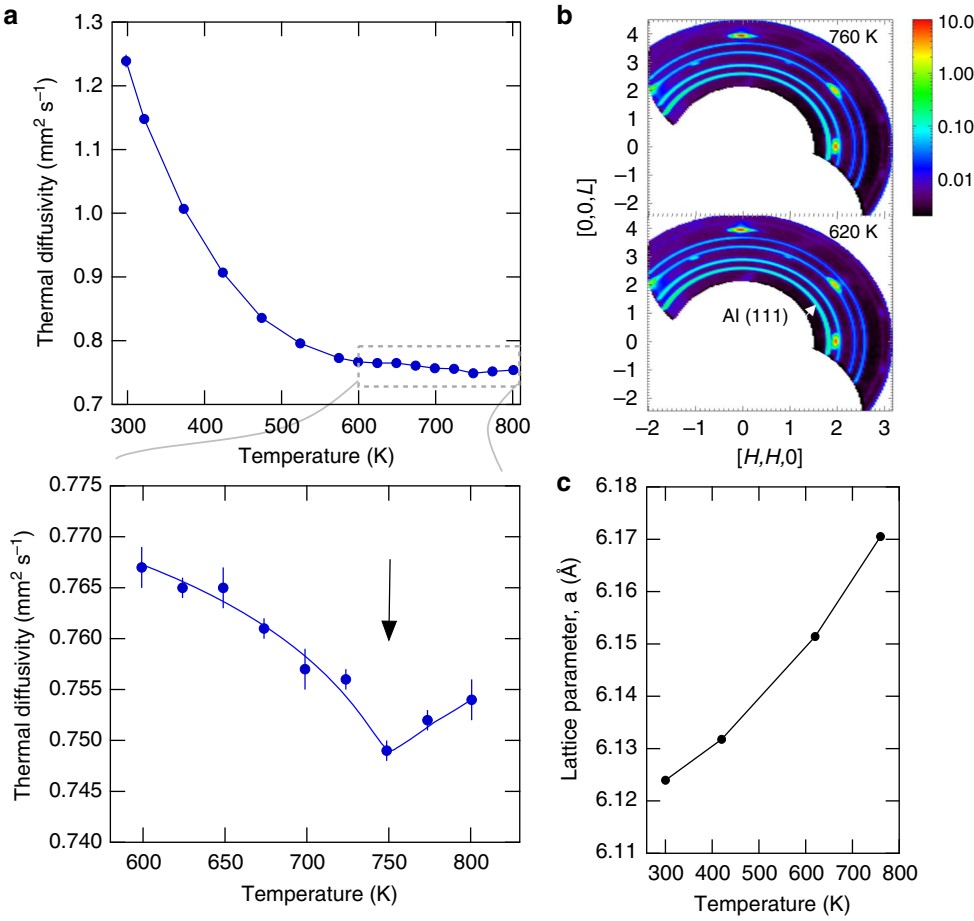

**Fig. 5** Thermal diffusivity anomaly absent a structural transition. **a** Thermal diffusivity measured using laser flash method on a piece of the same crystal used in the neutron scattering measurements. Lower panel shows a closeup view of a small negative kink in the data near 750 K (line is guide to the eye). **b** Single crystal diffraction pattern measured on the HYSPEC instrument above and below the temperature of the kink. The rings are aluminum powder rings from the crystal holder, which were used to calibrate the instrument when determining the PbSe lattice parameter. **c** Lattice parameter as a function of temperature determined by fitting the single crystal diffraction peaks. The refined lattice parameter of 6.124 ± 0.0005 Å at 300 K is in good agreement with the known value (the Al rings were used to calibrate the instrument assuming an Al lattice parameter of 4.046 Å at room temperature)

measurements and found no new Bragg reflections across the transition (Fig. 5b) and the temperature dependence of the deduced lattice parameter appears consistent with thermal expansion (Fig. 5c). Taken together, the ILM feature (Fig. 2), the flattening and fragmenting of the TO phonon (Figs. 1 and 2), and the corresponding dip in the thermal diffusivity (Fig. 5a) are all consistent with a transition in the anharmonic dynamical pattern similar to that observed in NaI[28]. This interpretation ties the TO phonon flattening to the ILM and underlying anharmonic dynamics.

Previous thermal conductivity measurements on PbSe poly-crystals do not show a clear kink but do show a plateau and in some cases a slight upturn above 750 K[19,30]. This makes sense if we consider that even small residual strains between grains can smear out the transition temperature if the energies involved are small enough to be comparable to variations in the local elastic strain energy[31]. Because the objective was to isolate intrinsic localization from impurity/defect localization, care was taken to maintain high purity (see Methods) and consequently our PbSe was not optimally doped for thermoelectric performance, having a maximum $zT \sim 0.45$ (Supplementary Fig. 3). Interestingly, for high-performance PbSe thermoelectrics with vacancy-induced dislocations there is instead an anomalous downturn in the thermal conductivity above ~750 K that results in a significantly

enhanced figure of merit[19]. An intriguing possible explanation for this could be that scattering of the LA phonon by dislocations compensates for the decrease in phonon–phonon scattering rates caused by the loss of phase space (Figs. 3 and 4), leaving only the phonon blocking effect of localization.

## Discussion

Our results reveal anharmonic localization in PbSe both in the form of an ILM feature and a transition to a flattened and fragmented TO phonon that we attribute to indirect localization resulting from an anharmonic pattern. Burlakov[25] derived anharmonic patterns in discrete lattices theoretically from an interference of extended mode instabilities and then noted that the resulting pattern can be regarded as a lattice of ILMs. Some challenges remain in fully reproducing such patterns from our ab initio simulations and there appears to be a difference in the temperature between theory and experiment. Nevertheless, the most striking features of the experiments, including the ILM, and the fragmenting and flattening of the TO phonon, are present in the simulation.

While anharmonic localization halts the propagation of the TO phonon, which reduces thermal conductivity, the doubling of the lifetime of the LA phonon increases thermal conductivity. Previous ab initio simulations[10] indicate that localization cuts the LA

contribution by about half by blocking the LA mode at the ILM energy. On the other hand, the doubling of the LA phonon lifetime at lower frequencies compensates for this in the pure crystal. However, our simulations indicate that this increased lifetime of the LA mode is a coincidence of how the phase space for scattering processes becomes reconfigured as localization develops. Hence, these anharmonic mechanisms are not optimized in PbSe to achieve the lowest possible thermal conductivity and consequently highest thermoelectric performance. A search for materials with a more optimal combination of localization and scattering should be explored both computationally and experimentally to understand all microscopic mechanisms controlling transport properties. As our results show, anharmonic interactions of strength beyond perturbative magnitudes can lead to fundamentally changed lattice transport properties.

## Methods

**PbSe single-crystal synthesis and characterization.** Single crystalline PbSe was prepared in a two-step process using Pb and Se materials with 99.999% metals-based purity. First, polycrystalline PbSe was obtained by reacting Pb and Se in a quartz ampoule at 1100 °C. The resulting material was crushed and placed into a quartz ampoule with a tapered bottom, which had been prepared by washing with nitric acid, rinsing with deionized water, and drying. A large single crystal (>10 g) was grown using the gradient freeze method. During this crystal growth, the quartz ampoule was supported by a quartz rod, which resulted in a cold finger effect that caused a crystal to be grown at the top of the ampoule due to vapor transport. This additional crystal proved useful because it cleaved easily, allowing the small needles utilized in inelastic x-ray measurements to be obtained and sealed under argon. The Hall effect is a natural probe of the defect levels in PbSe. Hall effect measurements were performed using a Quantum Design Physical Property Measurement system with maximum applied fields of ±6 T, and electrons were found to be the free carriers (Hall coefficient $R_H < 0$). The Hall carrier density, $n_H = 1/R_H e$ where $e$ is the fundamental charge, was ≈$2 \times 10^{17}$ cm$^{-3}$ and $2 \times 10^{18}$ cm$^{-3}$ for pieces of the main ingot and vapor transport crystal, respectively. X-ray analysis showed that the large single crystal had a low-angle boundary (~3°) along one edge, and this was removed by cutting this section away with a diamond saw. The larger remaining 8-g single crystal was used for neutron scattering.

**Triple-axis inelastic thermal neutron scattering.** The 8-g single crystal of PbSe was measured using the BT7 triple-axis spectrometer at the NIST Center for Neutron Research[32]. The spectrometer was operated with filtered fixed final neutron energy of 14.7 meV with horizontal collimation[33] 120′:80′:80′:120′, and the crystal was mounted in a furnace with the (HHL) reflections in the scattering plane using a vanadium holder. Vanadium is used for holders because of its incoherent neutron scattering cross-section and because it can withstand high temperatures. The instrument used pyrolytic graphite PG(002) for the monochromator and the analyzer. To measure the transverse optic mode near the zone center, where an ILM had been predicted[10], measurements were performed along $\mathbf{Q} = [H, H, 3]$ at 11 equally spaced points in $H$ from the (113) zone center to $H = 0.75$, which is half way to the zone boundary. The LA mode was measured along $\mathbf{Q} = [H, H, 0]$ at 15 equally spaced points in $H$ from the (220) zone center to $H = 2.35$. Scans were repeated at temperatures, $T = 300$, 643, and 793 K.

**Time-of-flight inelastic cold neutron scattering.** To look for any fine energy structure high-energy-resolution measurements were performed on the same 8 g crystal using the HYSPEC time-of-flight cold neutron spectrometer at the Spallation Neutron Source of Oak Ridge National Laboratory[34]. The crystal was again aligned in the (HHL) plane in a furnace using the same vanadium holder as with the triple-axis measurements. Measurements were performed at $T = 294$ and 770 K with an incident neutron energy of $E_i = 17$ meV and an energy resolution of $\Delta E \sim 0.3$ meV (FWHM) at 5 meV, the energy of the predicted sharp local mode feature[10]. A volume of data in $\mathbf{Q}$-$E$ space was obtained by rotating the angle between the [100] axis and the incident beam in 0.5° steps and combining the data using the HORACE software package[35]. The data were collected at each angle from −45° to +90° to obtain a complete data set. Additional counting was done in the range from −90° to −20° to obtain better statistics in the region of particular interest within the (113) zone.

**Inelastic x-ray scattering.** Additional phonon measurements were performed using IXS for the benefits of much smaller background and better out-of-plane $\mathbf{Q}$ resolution than the neutron scattering measurements. Measurements of the LA phonon were performed on a ~20 μm thick by 1 mm long splinter of the PbSe single crystal using the HERIX-30 X-ray spectrometer at beamline 30-ID-C at the Advanced Photon Source (APS)[36,37] with 23.7 keV ($\lambda = 0.5226$ Å) focused to a beam size of 35 μm × 10 μm. The ~20 μm thickness was chosen to maximize the scattering signal in transmission IXS measurements. The instrumental energy

resolution for IXS scans was 1.5 meV (FWHM), smaller than the measured linewidths. The PbSe single crystal splinter was loaded in a 100 μm ID glass capillary with 6 μm walls and sealed under 1/3 an atmosphere of argon as a high-temperature exchange gas. The capillary was then attached to a copper heater using silver paint and sealed under a beryllium dome for high temperature measurements. Measurements of the LA phonon were performed along $\mathbf{Q} = (2 + \varsigma, 2 + \varsigma, 0)$ at $T = 294$ and 770 K.

**Simulation methods.** First principles molecular dynamics simulations were carried out using the projector augmented-wave (PAW) method[38] as implemented in the Vienna Ab initio Simulation Package (VASP)[39–42], including polar effects on phonons[43]. Exchange and correlations were treated using the AM05 functional[44,45], and 600 eV was used as the plane-wave energy cutoff. The molecular dynamics were carried out for ~50,000 time steps of 2 fs each with a $5 \times 5 \times 5$ repetition of the unit cell (250 atoms), with temperature controlled by a Nose–Hoover thermostat[46,47]. The trajectories were analyzed with the temperature dependent effective potential method (TDEP)[10,29,48], extracting second and third order force constants from which the theoretical $S(\mathbf{Q}, E)$ was calculated[29,49,50].

The scattering function for coherent excitation creation was derived from the resulting momentum-resolved energy spectrum using[51]

$$S_{\text{coh}+1}(\mathbf{Q}, E) \propto \sum_{j, \mathbf{q}, \mathbf{G}} \frac{\left\langle n_j(\mathbf{q}) + 1 \right\rangle}{\omega_j(\mathbf{q})} |F(\mathbf{Q})|^2 \times \delta\left(E - \hbar\omega_j\right) \delta(\mathbf{Q} - \mathbf{q} - \mathbf{G}). \quad (1)$$

In this equation, upper case $\mathbf{Q}$ is the scattering wavevector, lower case $\mathbf{q}$ is the excitation wavevector, and $\mathbf{G}$ is a reciprocal-lattice wavevector. The factor $n_j(\mathbf{q})+1$ is the Bose–Einstein population $n_j = 1/\left(\exp\left[\frac{\hbar\omega_j}{kT}\right] - 1\right)$ plus one for phonon creation. The delta functions guarantee conservation of energy and momentum in the scattering process. The structure factor $F(\mathbf{Q})$ is given by

$$F(\mathbf{Q}) = \sum_d \frac{\overline{b_d}}{\sqrt{M_d}} \left(\mathbf{Q} \cdot e_d^j\right) \exp(i\mathbf{Q} \cdot \mathbf{r}) \exp(-W_d), \quad (2)$$

where the sum is over the atoms in the unit cell. The mass of the $d$th atom in the unit cell is $M_d$, $b_d$ is its neutron scattering length, $\mathbf{r}$ is its position vector, and exp $(-W_d)$ is the Debye–Waller factor. For each wavevector, $\mathbf{q}$, the atomic displacements for a given branch is characterized by the polarization vectors $e_d^j(\mathbf{q})$ for the atoms in the unit cell.

**Thermoelectric properties.** Thermal diffusivity was measured using a NETZSCH LFA457 laser flash system. It follows ASTM E1461[52]. The system uses an Nd-YAG laser to deposit a short heat pulse (0.1–1.0 ms) to heat up the front surface of a sample and an InSb IR detected measures the back-surface temperature rise. Thermal diffusivity is calculated using Cowan's method[53] with pulse-width correction. Thermal diffusivity was measured from 300 to 800 K in 50 K increments under argon purge gas at 100 ml/min. At each set point, three measurements were carried out. Thermal diffusivity results can be used to calculate thermal conductivity using: $\kappa = \alpha\rho C_P$ in which $\alpha$ is thermal diffusivity, $\rho$ is density, and $C_P$ is the specific heat[54].

Seebeck coefficient and electrical resistivity of the PbSe crystal was measured using a ULVAC-Riko ZEM-3 system. The measurement was carried in −0.09 MPa static helium after 3 cycles of helium gas purge and rotary pump evacuation. The sample was measured from 323 to 578 K in 50 K increments and from 578 to 794 K in 25 K increments. The sample temperature is determined by the average of the two R-type thermocouple probes in contact with the specimen. At each set point, resistivity was measured before temperature gradients were applied. Three temperature gradients at 10, 15, and 20 °C were applied to the bottom of the sample in order to determine Seebeck coefficient. Details of the measurement, system calibration, and accuracy can be found in ref. [55].

## Data availability

The data that support the findings of this study are available from the corresponding author on request.

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

## Acknowledgements

The authors thank D. Bansal for assistance in orienting the PbSe crystal. This work was supported by the US Department of Energy, Office of Science, Office of Basic Energy Sciences, Materials Sciences and Engineering Division under Contract Number DE-AC05-00OR22725. A portion of this research performed at the Oak Ridge National Laboratory's Spallation Neutron Source was sponsored by the US Department of Energy, Office of Basic Energy Sciences. The authors acknowledge the support of the National Institute of Standards and Technology, US Department of Commerce, in providing the neutron research facilities used in this work. The identification of any commercial product or trade name does not imply endorsement or recommendation by the National Institute of Standards and Technology. This research used resources of the Advanced Photon Source, a U.S. Department of Energy (DOE) Office of Science User Facility operated for the DOE Office of Science by Argonne National Laboratory under Contract No. DE-AC02-06CH11357. H. Wang's effort was sponsored by the DOE Energy Efficiency and Renewable Energy, Office of Vehicle Technologies Materials program. N.S. and A.J.M. acknowledge the support of the DARPA MATRIX program under Grant No. HR0011-15-2-0039. This work used the Extreme Science and Engineering Discovery Environment (XSEDE), which is supported by National Science Foundation Grant No. ACI-1053575.

## Author contributions

M.E.M. designed the experiments with theoretical guidance from A.J.M., N.S., and O.H. The triple-axis thermal neutron scattering measurements were performed by M.E.M., P.J.S. and J.W.L. The time-of-flight cold neutron scattering measurements were performed by V.O.G., N.S., and M.E.M. The inelastic x-ray scattering measurements were performed by A.A., M.E.M., R.P.H. and J.D.B. The neutron and x-ray scattering data were analyzed by M.E.M. The ab initio simulations were performed by O.H., N.S., and A.J.M. The single crystal growth and Hall effect measurements were performed by A.F.M. and B.C.S. The thermal diffusivity, Seebeck coefficient, and electrical resistivity measurements were performed by H.W. The manuscript was written by M.E.M. with input from all authors.

## Additional information

**Competing interests:** The authors declare no competing interests.

