## [Peer Review File · Nature Communications]

Reviewers' comments: Reviewer #1 (Remarks to the Author):

Review summary for Intrinsic anharmonic localization in thermoelectric PbSe

In this manuscript, the authors conducted inelastic neutron and x-ray scattering experiments on PbSe to study the effect of anharmonicity on the temperature dependence of the phonon dispersion. The results of the neutron scattering measurements show evidence of an intrinsic localized mode in the transverse optical mode at high temperature (at approx. 800 K). Surprisingly, linewidths obtained from x-ray scattering experiments indicate that the lifetime of longitudinal acoustic modes increases with increasing temperature. Based on simulations, the authors attribute these results to a change in the scattering phase space due to localized intrinsic modes.

On the whole, the study is meticulous and the conclusions seem well supported by the data. I would recommend publication in Nature Communications after the authors address the following concerns:

Text Comment

Page 4

Our results reveal that localization occurs at temperatures close to predicted but is more complete than expected and drives unanticipated changes in the lattice dynamics.

The authors could expand in their claim of “unanticipated” changes

This observation of such in-band localization in a PbSe crystal not only expands the domain of anharmonic localization to include entire phonons it also has important ramifications

“Entire phonons” is not very clear here.

Do the authors mean entire phonon branches?

Page 6

However, the high-resolution measurement reveals an additional sharp but weak

dispersionless feature near 5.5 meV

(indicated by arrow in Fig. 1a).

I believe this is meant to refer to Fig 2a)

Figure Comment

Fig 1a Is there a reason for not including the dispersion of LA and TA at room temperature? Was it not measured?

Fig 3a Arrows indicating the position of LA for the other values of Q could be included for clarity.

Fig 4 The figure illustrates the phase space for 3-phonon scattering events. What isn't clear, is whether the surface represents the allowed values of q_2 , q_3 , or $q_2 - q_3$. Some additional explanation would be valuable.

In addition to the minor comments above, the paper would benefit overall from more discussion of the origin (or potential origin) for the splitting off of part of the TO mode at high temperatures. What would cause such an effect, what is the expected impact on transport properties, etc.? Without such discussion, it is difficult to appreciate the impact of the work as a whole.

Reviewer #2 (Remarks to the Author):

Generally, thermoelectric materials attracting significant attention which is reflected also in numerous high impact papers. In this manuscript the authors present phonon dispersion curves of a defect-free PbSe single-crystal measured by inelastic neutron- and X-ray scattering. This manuscript describes new experimental findings, which go even beyond theoretical expectations and could provide a new strategy for thermoelectric material design. Like that, publication in Nature Communications could be deserved. Some of the authors published earlier simulation results which could quantitatively reproduce thermal conduction of PbSe and other related materials, where others failed even qualitatively. Their calculation predicts the presence of an isolated localized mode (ILM), which was indeed found experimentally here for the first time in this context (earlier observed for NaI). The highlight is however the discovery of a (broad) optical phonon branch with nearly zero group velocity at 793K (Fig. 1b), which decreases the thermal conductivity. At the same time however, the LA mode sharpen, i.e. its lifetime increases. The overall effect on thermoelectric performance is not known, unfortunately. Furthermore, the authors do not take into account important aspects and so their explanation is neither well founded nor has a predictive power. Therefore, I recommend a rejection. Nevertheless, I encourage the authors to improve their manuscript and depending on the amount of changes they should either resubmit it or submit it to another journal. A strategy could be to skip the high temperature part (which is fairly incomplete and unexplained), concentrate on the experimental evidence of the ILM now, and try to publish it somewhere else. The high temperature part alone (which seems not to be directly linked to the ILM) has a groundbreaking potential, but that should be proven first. Presenting the ILM only is a nice story already, which completes the computational work and explains the "low temperature" features. The localized optical mode is something very new, and exciting, which pushes the ILM story in the back unfortunately. However that story is not yet strong enough.

Major issues:

- The figures show the observed localized optical mode only in part of the Brillouin zone, therefore its impact is not clear. Furthermore, the presence of dispersionless optical mode is linked to the longer lifetime of LA phonons, which causes opposite effect on the thermoelectric performance. The authors claim that in other materials a better combination is possible. Nevertheless the authors should estimate the overall impact (semi)-quantitatively, if possible.
- The manuscript handles solely lattice dynamics, without taking electronic properties seriously into account, which is equally important for thermoelectric performance. According to present knowledge in the field, the electronic properties not only changes the electronic conduction part in the thermoelectric figure of merit (as mentioned in the manuscript), but also directly influence the lattice dynamics through electron-phonon coupling. It has been shown for SnSe that indeed phonon anharmonicity is causing the low thermal conductivity, which arises from electron-phonon interactions [C.W. Li et al.: Nature Physics 11 (2015) 1063-1069]. Please see also in general: W. G. Zeier et al.: Angew. Chem. Int. Ed. 55 (2016) 6826-6841
- The localized mode appears quite suddenly (the spectrum at 300 and 643 K do not display that feature: Fig. 1b vs. 1c and d.) It has been reported that SnSe and SnS undergoes a phase transition at a similar temperature [Y. K. Lee et al.: J. Am. Chem. Soc. 139 (2017) 10887-10896 & T. Chattopadhyay et al.: Revue Phys. Appl. 19 (1984) 807-813]. The sudden change in the phonon

spectrum suggest a phase transition also for PbSe, although I did not find any report on that. Therefore, the authors should prove the structure as a function of temperature experimentally, which can be a key puzzle piece. Furthermore, they

- It is also known, that with increasing temperature the off-centering of Pb^{2+} tends to increase due to the s_2 lone pair of Pb^{2+} . The authors point out in their previous publication that their simulation does not show any off-centering. It can't be excluded without experiments that off-centering happens in reality, and that the calculation needs to be extended to capture this feature. In fact, a phase transition and the observed localized optical mode can be connected to that. The calculated phonon spectra is based on the temperature dependent effective potential method. The authors have shown in an earlier paper [their ref. 8] that this approach is crucial for anharmonic materials and that it is able to quantitatively reproduce the thermal conductivity of PbS, PbSe and PbTe. But the agreement is shown only up to 700K, which is likely to be below the temperature, where the localized mode appears. The authors should prove that their calculation of heat conductivity is reliable up to at least 800K, where the optical mode gets localized.

Related to this: Wang et al. [Y Wang et al: npj Computational Materials (2016) 16006] pointed out that it can be important to take the polar nature of the material into account. According to their table, VASP is not able to do that. Maybe this could be a missing piece?

Minor comments:

- It is well known that thermoelectric performance of PbSe (and of course also others) depends strongly on charge carrier density. The authors have chosen the approach to study defect free single crystal to eliminate this effect and study the lattice dynamics in a good crystal. It is expected that the thermoelectric performance of their crystal is not spectacular. Since the observed localization of the optical phonon branch is not explained, the question remains, whether the same effect can be expected if other measures are simultaneously applied to increase the thermoelectric performance.

- P-C. Wei et al. [Nature 539, E1–E2] points out that extensive and precise characterization of the related SnSe crystals are prerequisite for reproducibility and for drawing correct conclusions. Please give detailed information on sample preparation, characterization in the SI (e.g. density, in comparison to the theoretical one; structure refinement for all temperatures – and compare it to the structure obtained from the simulation; finally, as the most important for thermoelectric materials, please provide zT for all relevant temperatures)

Details:

- Title: "Intrinsic anharmonic localization in thermoelectric PbSe" and in the Abstract: "Our study shows how strong anharmonicity can fundamentally alter vibrational transport properties even in perfect crystals." Is the small ILM or the optical mode meant in the Title? I assume the later one. It is not shown that the localized optical mode is caused by anharmonicity. This is neither predicted by the calculations, nor argued sufficiently.

- Abstract: "zero group velocity develops for the full spectral weight of transverse optic (TO) phonon. Arrest of TO phonon propagation..." These statements are misleading, the localised mode is shown only for a small part of the Brillouin zone.

- Abstract: "coincides with unusual sharpening of the longitudinal acoustic mode due to a loss of phase space for phonon scattering." Sta

- p.4.: "Our results reveal that localization occurs at temperatures close to predicted" It can be, but not shown. Ref. 8. does not state a clear temperature(range) where ILM gets separated from

TO. It is also not shown, how sudden is that change.

- Fig. 1.b: Can the uncommented branch at about 3 meV be ILM?

- "As shown in Fig. 1e,f a large anomalous decrease in the energy linewidths for the LA phonon measured along $Q = [H, H, 0]$ with heating from 643 K to 793 K"

give fit details (parameters and restrictions) together with resolved fit spectra in SI, also for Fig.3.

-Error bars should be either (visible) plotted or described in the figure caption.

- "relaxed out-of-plane angular (or Q) resolution of the BT7" what is the effect on the other branches? Can it be excluded that the ILM is an artefact?

Ideally, a second picture of the calculated dispersion curves should be presented additionally (one of them in the SI), which includes all (Q and E) resolution effects.

- "The TA mode also appears, but is weaker than with the triple axis measurements (Fig. 1e,f) because of a better out-of-plane Q resolution." Is this true quantitatively? IXS should have much better Q resolution, so it is surprising to still see TA.

- A weak point of the work is the interaction with other researchers. Important references are missing, like:

recent review on PbSe: C. Gayner et al.: Materials Today Energy 9 (2018) 359-376

review about strategies for thermoelectric materials: M.Beekman et al.: Nature Materials 14 (2015) 1182-1185

doping effect on zT in PbSe: H. Wang et al.: PNAS 109/25 (2012) 9705-9709

- Please provide Grüneisen parameter(s), if possible both experimental and calculated ones.

- Methods/TAS INS: Please give details of data collection plan and comment it.

- Methods/ToF INS: Please give details to data reduction, e.g. software used, major steps done.

Fanni Juranyi

Reviewer #3 (Remarks to the Author):

This work reported experimental observation of phonon localization at elevated temperature using neutron scattering technique. The observed localization applied to the full spectrum of transverse optic mode, which is different than the theoretical prediction. In addition, sharpening of the longitudinal acoustic mode was discovered. These observations provide deeper understanding of the phonon transport in an important thermoelectric material, PbSe.

This work is worth publishing should the following comments be addressed properly

1) The authors should report result on impurity measurement, and discuss potential effect on phonon scattering; and 2) The authors should provide data on thermal conductivity or diffusivity.

Minor issues found in the manuscript

3) Figure 1a should include visible error bars

4) In line #7 of paragraph 2 on page 6, the citation of the figure should be "Fig. 2a".

Reviewer #1

Comment 1 of Reviewer 1: *In this manuscript, the authors conducted inelastic neutron and x-ray scattering experiments on PbSe to study the effect of anharmonicity on the temperature dependence of the phonon dispersion. The results of the neutron scattering measurements show evidence of an intrinsic localized mode in the transverse optical mode at high temperature (at approx. 800 K). Surprisingly, linewidths obtained from x-ray scattering experiments indicate that the lifetime of longitudinal acoustic modes increases with increasing temperature. Based on simulations, the authors attribute these results to a change in the scattering phase space due to localized intrinsic modes.*

On the whole, the study is meticulous and the conclusions seem well supported by the data. I would recommend publication in Nature Communications after the authors address the following concerns:

Reply: We thank the reviewer for this summary and positive assessment of our manuscript.

Comment 2 of Reviewer 1:

Page 4	
Our results reveal that localization occurs at temperatures close to predicted but is more complete than expected and drives unanticipated changes in the lattice dynamics.	The authors could expand in their claim of “unanticipated” changes

Reply: In our revised manuscript we have added some additional details related to unanticipated changes. First, we mention the unexpected sharpening of the LA phonon at high temperatures. This sentence now reads, “*Our results reveal that localization occurs at temperatures close to predicted but is more complete than expected and drives unanticipated changes in the lattice dynamics, including an unexpected sharpening of the longitudinal acoustic phonon at high temperatures.*” Then we expanded upon the description of the more interesting aspects of the observed localizations, “*Rather than localization occurring with a fraction of the intensity of the normal phonons, as predicted¹⁰, the entire spectral weight of a large portion of the transverse optic phonon abruptly develops flat dispersion (zero group velocity) and appears fragmented. The localization (flattening) and fragmenting of the optic phonon is explained in terms of a transition in the anharmonic dynamics^{27, 28}, which is also detected as a small kink in our thermal diffusivity measurements similar to that observed with ILM ordering in NaI²⁸.*” Finally, to tie the thought back to the LA phonon sharpening we add the ending phrase to this sentence, “*We find that the rearrangement of spectral features that comes with localization also fundamentally changes the phase space for scattering, which explains the sharpening of the LA phonon.*”

Comment 3 of Reviewer 1:

This observation of such in-band localization in a PbSe crystal not only expands the domain of anharmonic localization to include entire phonons it also has important ramifications	“Entire phonons” is not very clear here. Do the authors mean entire phonon branches?
--	--

Reply: Normally ILMs just steal a small part of a phonon’s spectral weight and we were just trying to express the fact that here all the weight of the phonon goes flat in a part of the zone. We can now see how this could be misleading. We have deleted the phrase “**to include entire phonons**” from this sentence in the revised manuscript. In the abstract we also clarify this point by noting, “- *zero group velocity develops abruptly for a significant section of the transverse optic (TO) phonon...*”

Comment 4 of Reviewer 1:

Page 6	
However, the high-resolution measurement reveals an additional sharp but weak dispersionless feature near 5.5 meV (indicated by arrow in Fig. 1a).	I believe this is meant to refer to Fig 2a)

Reply: We have corrected this error.

Comment 5 of Reviewer 1:

Fig 1a	Is there a reason for not including the dispersion of LA and TA at room temperature? Was it not measured?
---------------	---

Reply: Yes, this scan was not measured at room temperature during the triple axis measurements. When these experiments began the main focus was on observing the predicted localization in the TO phonon. The sharpening of the LA phonon at high temperatures took us by surprise. With the BNL furnace we were using (<https://www.nist.gov/ncnr/sample-environment/sample-environment-equipment/furnaces/bnl-furnace>) it was not practical to return to room temperature because this furnace has no cooling power (it is meant for 200 - 600°C but room temperature is possible before the heater is turned on). Rather than returning to NIST for another measurement, we thought it made more sense to observe the same behavior using IXS, which has the advantage of better momentum resolution and lower background.

Comment 6 of Reviewer 1:

Fig 3a	Arrows indicating the position of LA for the other values of Q could be included for clarity.
---------------	---

Reply: Arrows for the other values of Q are now included in revised Fig. 3a.

Comment 7 of Reviewer 1:

Fig 4	The figure illustrates the phase space for 3-phonon scattering events. What isn't clear, is whether the surface represents the
	allowed values of q_2 , q_3 , or q_2-q_3 . Some additional explanation would be valuable.

Reply: The surface represents allowed values for \mathbf{q}_2 for a fixed \mathbf{q}_1 ($\mathbf{q}_3 = -\mathbf{q}_1 - \mathbf{q}_2$ is not shown). This clarification is now made in the Figure caption. There was also an error in the labeling of the indicies that has been corrected in the text.

Comment 8 of Reviewer 1: *In addition to the minor comments above, the paper would benefit overall from more discussion of the origin (or potential origin) for the splitting off of part of the TO mode at high temperatures. What would cause such an effect, what is the expected impact on transport properties, etc.? Without such discussion, it is difficult to appreciate the impact of the work as a whole.*

Reply: In our revised manuscript we have expanded the discussion and analysis of anharmonic localization with additional experiments, references, and calculations. The simplest explanation for a local vibration breaking from the main phonon is the ILM concept. The ILM concept is the idea that a local amplitude fluctuation shifts the vibrationally frequency sufficiently because of nonlinearity that it breaks away from the normal modes. However, as Burlakov showed interference of mode instabilities in an anharmonic lattice can also produce dynamical patterns that can be described as lattices of ILMs (Burlakov, V. M. Interference of mode instabilities and pattern formation in anharmonic lattices. *Phys. Rev. Lett.* **80**, 3988 (1998). <https://journals.aps.org/prl/pdf/10.1103/PhysRevLett.80.3988>). We now include this reference and mention patterns in the introduction, results section, and as part of the discussion.

The formation of dynamical patterns has interesting consequences; one of which is that other modes will also tend to localize as they become trapped between the ILMs. An array of ILMs introduces an array of soft spots in the lattice. This effect was illustrated for NaI with a simple model where the force constant on every 4th light atom is softened, see Fig. 5 in <https://www.nature.com/articles/srep00004>. In addition to a local mode the optic phonon both flattens and fragments. Following the work on NaI, we performed additional measurements to test whether a transition was occurring in the dynamical pattern, which would help explain why the TO phonon abruptly flattens and fragments at high temperatures. Indeed, we find evidence for this in thermal diffusivity measurements in the form of a kink with no corresponding change in the crystal structure, see revised manuscript Fig. 5 and our more detailed response to *Reviewer 2* below. We also show with additional *ab initio* simulations that a similar effect occurs in the calculations but requires a somewhat higher temperature, see revised Fig. 2e.

The *ab initio* simulation indicates that the optic and acoustic modes contribute about equally to the total lattice thermal conductivity of PbSe (Ref. [10], Shulumba *et al.* *Phys. Rev. B* **95**, 014302 (2017)). The LA phonon makes up about a quarter of the total. The low overall thermal conductivity is driven by strong anharmonic interactions between the TO and LA phonons, from

which the ILM results. At the ILM energy the LA phonon contribution essentially vanishes, and the ILM itself does not propagate energy. In the spectral thermal conductivity plot this creates a gap in the LA contribution that amounts to a loss of about half of the spectral weight of the LA branch. On the other hand, the doubling of the LA phonon lifetime at lower frequencies found experimentally doubles the other part. Hence, these contributions tend to cancel each other.

As discussed in our reply to *Comment 1 of Reviewer 2* below, there is also published evidence suggesting that the balance might be tipped in favor of lower thermal conductivity and thus higher thermoelectric efficiency with the introduction of defects (dislocations) (see Chen, Z., *et al.* Vacancy-induced dislocations within grains for high-performance PbSe thermoelectrics. *Nat. Commun.* **8**, 13828 (2017)).

Reviewer #2

Comment 1 of Reviewer 2: *“Generally, thermoelectric materials attracting significant attention which is reflected also in numerous high impact papers. In this manuscript the authors present phonon dispersion curves of a defect-free PbSe single-crystal measured by inelastic neutron- and X-ray scattering. This manuscript describes new experimental findings, which go even beyond theoretical expectations and could provide a new strategy for thermoelectric material design. Like that, publication in Nature Communications could be deserved. Some of the authors published earlier simulation results which could quantitatively reproduce thermal conduction of PbSe and other related materials, where others failed even qualitatively. Their calculation predicts the presence of an isolated localized mode (ILM), which was indeed found experimentally here for the first time in this context (earlier observed for NaI). The highlight is however the discovery of a (broad) optical phonon branch with nearly zero group velocity at 793K (Fig. 1b), which decreases the thermal conductivity. At the same time however, the LA mode sharpen, i.e. its lifetime increases. The overall effect on thermoelectric performance is not known, unfortunately. Furthermore, the authors do not take into account important aspects and so their explanation is neither well founded nor has a predictive power. Therefore, I recommend a rejection. Nevertheless, I encourage the authors to improve their manuscript and depending on the amount of changes they should either resubmit it or submit it to another journal. A strategy could be to skip the high temperature part (which is fairly incomplete and unexplained), concentrate on the experimental evidence of the ILM now, and try to publish it somewhere else. The high temperature part alone (which seems not to be directly linked to the ILM) has a groundbreaking potential, but that should be proven first. Presenting the ILM only is a nice story already, which completes the computational work and explains the “low temperature” features. The localized optical mode is something very new, and exciting, which pushes the ILM story in the back unfortunately. However, that story is not yet strong enough.”*

Reply: We thank the reviewer for this summary and thoughtful critique; it has stimulated significant improvements to our manuscript. However, we think separating the ILM result from the optic mode localization is not the right approach because we believe these two effects are connected. Let us explain.

If the ILMs form as patterns in PbSe like they do in NaI (Manley *et al.* PRB **89**, 224106 (2014)) then the optic mode would tend to localize. The figure below demonstrates this using a diatomic

1-d lattice model, which is based on a calculation for NaI
<https://www.nature.com/articles/srep00004>. Creating an array of ILMs by softening the force constant on every 4th light atom we see that a local mode (LM) appears and the optic phonon both flattens and breaks up.

This is essentially a band folding effect resulting from the superstructure defined by the pattern of LMs. Whenever the optic mode wavevector matches a multiple of the LM pattern a standing wave results, which tends to flatten the optic band overall (indirect localization). Or, in other words, the optic modes become trapped between the LMs. Note that the localization of each new optic branch is not perfect, but the entire band looks flat and fragmented. This suggests that the sudden flattening and fragmenting of the TO phonon at 793 K is caused by an ILM pattern forming at high temperatures. Note that for ILMs the lattice softening is induced by the intrinsic anharmonicity.

Such ILM patterns forming in NaI are described in detail in

<https://www.nature.com/articles/srep00004>

and <https://journals.aps.org/prb/abstract/10.1103/PhysRevB.89.224106>

The first paper in *Sci. Rep.* shows how the patterns manifest in the inelastic spectrum, which includes the mentioned flattening of the TO phonon. The second paper in *Phys. Rev. B* shows that these transitions can also be detected as subtle changes in the thermophysical properties. Note that such anharmonic dynamical patterns can be described either as lattice of ILMs or an interference pattern of anharmonic modes (Burlakov, V. M. Interference of mode instabilities and pattern formation in anharmonic lattices. *Phys. Rev. Lett.* **80**, 3988 (1998)). The key point is that there is a pattern of soft spots in the crystal caused by anharmonic vibrations.

Because the enthalpy/entropy changes are small they can easily be missed in conventional scanning calorimetry methods, which tend to smear out small features, but the laser flash method was shown to be sensitive to these transitions. Therefore, we performed additional measurements on our crystal using a laser flash system. The thermal diffusivity measured on a piece of the same crystal used in the neutron measurements indeed shows a subtle kink in the thermal diffusivity at 750 K.

This feature at 750 K indicates a transition that is in the right temperature range to explain the flattening of the TO phonon at 793 K (Fig. 1 in manuscript).

Because such transitions are primarily from changes in the anharmonic dynamical pattern, changes in the static structure are small and can only be detected as slight inhomogeneous deviations from normal thermal expansion in high precision diffraction, see <https://journals.aps.org/prb/abstract/10.1103/PhysRevB.89.224106>. The structure of the PbSe indeed shows no obvious structural change across this temperature. The single crystal diffraction peaks appear unaltered, there are no new reflections, and the temperature dependence of the lattice parameter appears normal in the HYSPEC measurements:

These results taken together, the ILM feature (Fig. 2), the flattening and fragmenting of the TO phonon (Figs. 1&2), and the corresponding dip in the thermal diffusivity (Fig. 5a) are all consistent with a transition in the anharmonic dynamical pattern similar to that observed in NaI crystals. This interpretation provides an explanation for the flattening of the TO phonon and ties it to the ILM feature and underlying anharmonic dynamics. It also explains how substantial changes can develop in the lattice dynamics with minimal effects on the crystal structure.

Previous thermal conductivity measurements on PbSe polycrystals do not show a clear kink but do show a plateau and in some cases a slight upturn above 750 K. This makes sense if we consider that elastic strains between grains can smear out a transition when the transition enthalpy is small enough to be comparable to local elastic strain energies. Interestingly, for high-performance PbSe thermoelectrics with vacancy-induced dislocations there is instead an anomalous downturn in the thermal conductivity above ~ 750 K that results in a significantly enhanced figure of merit (Chen, Z., *et al.* Vacancy-induced dislocations within grains for high-performance PbSe thermoelectrics. *Nat. Commun.* **8**, 13828 (2017).). An intriguing potential explanation for this could be that scattering of the LA phonon by dislocations compensates for the decrease in phonon-phonon scattering rates caused by the loss of phase space (Fig. 3, 4) leaving only the phonon blocking effect of localization. In future work this idea could be checked directly by measuring the LA and TO phonons in these PbSe high-performance thermoelectrics.

The possibility that anharmonicity and discreteness could produce patterns of organized ILMs rather than isolated ILMs was theoretically demonstrated by Burlakov in 1998:

Burlakov, V. M. Interference of mode instabilities and pattern formation in anharmonic lattices. *Phys. Rev. Lett.* **80**, 3988 (1998). <https://journals.aps.org/prl/pdf/10.1103/PhysRevLett.80.3988>

Burlakov derived anharmonic patterns theoretically from an interference of extended mode instabilities and then noted that the resulting pattern can be regarded as a lattice of ILMs. Some challenges remain in understanding such patterns from the *ab initio* simulations. Nevertheless, the most striking features of the experiments, including the ILM, the fragmentation, and flattening of the TO phonon, are present in the simulation. Furthermore, as discussed more in our reply to *Comment 12 of Reviewer 2* below, additional flattening and fragmenting of the TO phonon occurs in the simulations when performed at somewhat higher temperature (1000 K), suggesting that at least some of the differences between experiment and theory relate to an offset in the temperature scale.

These arguments and figures have been incorporated into our revised manuscript. In the introduction after the description of an isolated ILM we added a statement about how anharmonicity can also result in more complex dynamical patterns/ordered arrays of ILMs and include reference to the Burlakov PRL mentioned above. In the last paragraph of the introduction we also mention that the localization (flattening) of the optic phonon can be explained by a transition in the dynamical pattern, which manifests as a small kink in our thermal diffusivity measurements similar to that observed for gap ILMs in NaI, and include references to the two NaI papers mentioned above (<https://www.nature.com/articles/srep00004> and <https://journals.aps.org/prb/abstract/10.1103/PhysRevB.89.224106>). We add a new results section “Thermal diffusivity and structure analysis”, where we describe the kink in the diffusivity measurements and the absence of an associated structural transition in the context of the changes observed in the TO phonon. This new section includes a new figure that displays the thermal diffusivity and diffraction data (Fig. 5), along with a brief discussion of how our results compare with some previous measurements on polycrystalline samples. We also added a paragraph in the discussion about the ILM patterns and the challenges remaining in explaining the remaining differences between simulation and experiment. A section was also added to the Methods section “Thermoelectric properties measurements”, which includes descriptions of new

thermal diffusivity, electrical resistivity, and Seebeck coefficient measurements (together these are also used to determine a figure of merit zT curve for our crystal, which can be found in the SI). The simulations are also extended to 1000 K in Fig. 2.

Comment 2 of Reviewer 2: *The figures show the observed localized optical mode only in part of the Brillouin zone, therefore its impact is not clear. Furthermore, the presence of dispersionless optical mode is linked to the longer lifetime of LA phonons, which causes opposite effect on the thermoelectric performance. The authors claim that in other materials a better combination is possible. Nevertheless, the authors should estimate the overall impact (semi)-quantitatively, if possible.*

Reply: In order to estimate the overall impact on thermal conductivity we leverage both experimental results and the *ab initio* simulations. As noted, our dispersion measurements are an incomplete sampling of reciprocal space (as is typically the case) whereas the *ab initio* simulation include a complete measure but miss some experimental details (Fig. 2).

First, the *ab initio* simulation indicates that the optic and acoustic modes contribute about equally to the total lattice thermal conductivity of PbSe (Ref. [9], Shulumba *et al. Phys. Rev. B* **95**, 014302 (2017)). The LA phonon makes up about a quarter of the total. The low overall thermal conductivity is driven by strong anharmonic interactions between the TO and LA phonons, from which the ILM results. At the ILM energy the LA phonon contribution essentially vanishes, and the ILM itself does not propagate energy. In the spectral thermal conductivity plot this creates a gap in the LA contribution that amounts to a loss of about half of the spectral weight of the LA branch. On the other hand, the doubling of the LA phonon lifetime at lower frequencies found experimentally doubles the other part. Hence, these contributions tend to cancel each other. However, the anomalies observed in our measured thermal diffusivity (new Fig. 5 in the revised manuscript) shows that the cancellation is actually not perfect.

As discussed in our reply to *Comment 1 of Reviewer 2* above, there is also published evidence suggesting that the balance might be tipped in favor of lower thermal conductivity and thus higher thermoelectric efficiency with the introduction of defects (dislocations) (see Chen, Z., *et al. Vacancy-induced dislocations within grains for high-performance PbSe thermoelectrics. Nat. Commun.* **8**, 13828 (2017)). Measurements of the lattice dynamics on crystals with similar vacancy-induced dislocations are needed to test this idea. We hypothesize that such measurements might reveal that the scattering of the LA phonon by dislocations prevents the lifetime gains associated with a loss of phase space, leaving only the phonon blocking effect of localization.

We added a few sentences in the discussion paragraph briefly explaining these estimates, thereby making the discussion more quantitative. The text in the new section on thermal transport measurements also includes additional discussion on thermal transport measured by us on a pure crystal and others on polycrystals with and without vacancy-induced dislocation defects.

Comment 3 of Reviewer 2: *The manuscript handles solely lattice dynamics, without taking electronic properties seriously into account, which is equally important for thermoelectric performance. According to present knowledge in the field, the electronic properties not only*

changes the electronic conduction part in the thermoelectric figure of merit (as mentioned in the manuscript), but also directly influence the lattice dynamics through electron-phonon coupling. It has been shown for SnSe that indeed phonon anharmonicity is causing the low thermal conductivity, which arises from electron-phonon interactions [C.W. Li et al.: Nature Physics 11 (2015) 1063-1069]. Please see also in general: W. G. Zeier et al.: Angew. Chem. Int. Ed. 55 (2016) 6826-6841

Reply: We agree with the general point that our work focuses solely on the lattice dynamics, rather than the equally important electronic properties, and that anharmonicity is causing the low thermal conductivity. The case of SnSe is indeed interesting, but this system is quite different than PbSe. The PbSe, PbTe, and PbS systems are in the cubic rock-salt structure while SnSe is a layered orthorhombic structure (https://www.webelements.com/compounds/tin/tin_selenide.html).

The purpose of our experiments was to isolate the contributions of anharmonic phonon interactions as much as possible. Optimization of thermoelectric properties was intentionally avoided and not just because of the influence of electron-phonon coupling, but also because the impurities themselves could complicate the interpretation of the lattice dynamics owing to the fact that they can produce localized impurity modes.

To address the thermoelectric properties of our pure undoped crystal we now include measurements of thermoelectric properties in the revised manuscript and SI, including thermal diffusivity, electrical resistivity, Seebeck coefficient, and figure of merit zT for all relevant temperatures. The peak zT for our crystal is about 0.45.

We are aware of the paper by C.W. Li et al.: *Nature Physics* **11**, 1063-1069 (2015) and agree with its conclusions. This work was actually done in our group at ORNL, and we use similar experimental and theoretical approaches in the present work, but the physics is different. In particular, the intrinsic anharmonic localization opens an interesting new direction.

Comment 4 of Reviewer 2: *The localized mode appears quite suddenly (the spectrum at 300 and 643 K do not display that feature: Fig. 1b vs. 1c and d.) It has been reported that SnSe and SnS undergoes a phase transition at a similar temperature [Y. K. Lee et al.: J. Am. Chem. Soc. 139 (2017) 10887-10896 & T. Chattopadhyay et al.: Revue Phys. Appl. 19 (1984) 807-813]. The sudden change in the phonon spectrum suggest a phase transition also for PbSe, although I did not find any report on that. Therefore, the authors should prove the structure as a function of temperature experimentally, which can be a key puzzle piece.*

Reply: In our revised manuscript we now include a new figure, Fig. 5, that addresses this question. There is no evidence of a structural transition in the sense that the single crystal diffraction pattern is unaltered, and the lattice parameter follows normal thermal expansion behavior. However, thermal diffusivity shows a subtle kink at near 750 K. This is consistent with transitions observed in the dynamical structure of NaI in which the ILMs reorganize into a different pattern (<https://journals.aps.org/prb/abstract/10.1103/PhysRevB.89.224106>). An important consequence of ILM pattern formation is that it also tends to both fragment and flatten the optic phonon branch (<https://www.nature.com/articles/srep00004>). Hence, we can

understand why the TO phonon appears to both flatten and fragment suddenly when heating above 750 K, and the ILM and the more general localization of the TO phonon originate from the same anharmonic dynamics.

Comment 5 of Reviewer 2: *It is also known, that with increasing temperature the off-centering of Pb²⁺ tends to increase due to the s² lone pair of Pb²⁺. The authors point out in their previous publication that their simulation does not show any off-centering. It can't be excluded without experiments that off-centering happens in reality, and that the calculation needs to be extended to capture this feature. In fact, a phase transition and the observed localized optical mode can be connected to that. The calculated phonon spectra is based on the temperature dependent effective potential method. The authors have shown in an earlier paper [their ref. 8] that this approach is crucial for anharmonic materials and that it is able to quantitatively reproduce the thermal conductivity of PbS, PbSe and PbTe. But the agreement is shown only up to 700K, which is likely to be below the temperature, where the localized mode appears. The authors should prove that they calculation of heat conductivity is reliable up to at least 800K, where the optical mode gets localized.*

Reply: We are not convinced that *static* Pb off-centering occurs with increasing temperature in these systems. Although this argument has been made based on pair distribution function (PDF) measurements, separate analysis by independent research groups has shown convincingly that the Pb do not in fact off center, but rather that the asymmetry in the measured PDF is actually consequence of the anharmonic vibrations. Here are quotes from two papers we find convincing:

“Our first-principles calculations of the radial distribution function in both SnTe and PbTe show a clear asymmetry in the first nearest-neighbor (1NN) peak, which increases with temperature, in agreement with recent experimental reports. We show that this peak asymmetry for the 1NN Sn-Te or Pb-Te bond results from large-amplitude anharmonic vibrations (phonons). No atomic off centering is found in our simulations.”

C. W. Li *et al.*, *Phys. Rev. B* **90**, 214303 (2014).

<https://journals.aps.org/prb/abstract/10.1103/PhysRevB.90.214303>

“... we resolve an experimental controversy by showing that there are no appreciable local nor global spontaneously broken symmetries at finite temperature and that the anomalous spectral features simply arise from two anharmonic interactions.”

Chen, Y., Xinyuan, A. & Marianetti C. A. *Phys. Rev. Lett.* **113**, 105501 (2014).

<https://journals.aps.org/prl/abstract/10.1103/PhysRevLett.113.105501>

The point is that the anomalies in the pair distribution function are real, but they are derived from the anharmonic vibrations, not structural changes. This is also the case for intrinsic localized modes. They are first and foremost a consequence of the anharmonic dynamics (see Campbell, D. K., Flach, S. & Kivshar, Y. S. Localizing energy through nonlinearity and discreteness. *Phys. Today* **57**, 43–49 (2004)). Any subtle structural changes associated with them are secondary. This is why we can see fairly large and obvious changes in the dynamics, e.g. localization of the TO phonon in the measurements or a strong ILM in the simulations, while structural changes are not found in either the *ab initio* calculations or structural measurements.

We agree that there is a transition, but it is from changes in anharmonic dynamics (the dynamical pattern), not in the static structure.

We now include transport measurements that extend to higher temperatures (new Fig. 5). The temperature dependence of the thermal diffusivity tends to flatten out above about 600 K in the measurements. In the calculations the thermal resistivity (inverse) of PbSe also tends to flatten at high temperatures rather unlike PbTe and PbS which do not show localization.

Comment 6 of Reviewer 2: *Related to this: Wang et al. [Y Wang et al: npj Computational Materials (2016) 16006] pointed out that it can be important to take the polar nature of the material into account. According to their table, VASP is not able to do that. Maybe this could be a missing piece?*

Reply: The polar nature of the material is taken into account in our simulations. The implementation of polarization in VASP is described here,

M. Gajdoš, *et al.* Linear optical properties in the projector-augmented wave methodology *Phys. Rev. B* **73**, 045112 (2006).

<https://journals.aps.org/prb/abstract/10.1103/PhysRevB.73.045112>

which is a modification of this earlier paper:

Stefano Baroni and Raffaele Resta, *Ab initio* calculation of the macroscopic dielectric constant in silicon

Phys. Rev. B **33**, 7017 (1986).

<https://journals.aps.org/prb/abstract/10.1103/PhysRevB.33.7017>

Numerical treatment of polar effects on phonons is described in great detail here,

Gonze X. and Changyol Lee, Dynamical matrices, Born effective charges, dielectric permittivity tensors, and interatomic force constants from density-functional perturbation theory. *Phys. Rev. B* **55**, 10355 (1997).

<http://link.aps.org/doi/10.1103/PhysRevB.55.10355>

We use the treatment described by Gonze *et al.*, which is fast and correct. To indicate this in our revised manuscript we have added the reference by Gonze *et al.* in “Simulation methods” after the added phrase “...*including polar effects on phonons*”.

Minor comments:

Comment 7 of Reviewer 2: - *It is well known that thermoelectric performance of PbSe (and of course also others) depends strongly on charge carrier density. The authors have chosen the approach to study defect free single crystal to eliminate this effect and study the lattice dynamics in a good crystal. It is expected that the thermoelectric performance of their crystal is not spectacular. Since the observed localization of the optical phonon branch is not explained, the*

question remains, whether the same effect can be expected if other measures are simultaneously applied to increase the thermoelectric performance.

Reply: In the revised manuscript we now explain the localization of the optical phonon in terms of a transition in the anharmonic dynamical pattern (or ILM lattice) and find supporting evidence for it in thermal diffusivity measurements (Fig. 5).

We do not expect the localization to be disrupted by a small concentration of impurities. One reason is that in the case of NaI, where a similar localization occurs, results were the same for both pure NaI and NaI doped with 0.2% TI (a scintillator crystal). Even the ILM pattern formation temperature was the same for both doped and undoped NaI crystals.

For high-performance PbSe thermoelectrics with vacancy-induced dislocations there is an anomalous downturn in the thermal conductivity above ~750 K that results in a significantly enhanced figure of merit (see figure 1 in <https://www.nature.com/articles/ncomms13828>). An intriguing possible explanation for this could be that scattering of the LA phonon by dislocations compensates for the decrease in phonon-phonon scattering rates caused by the loss of phase space (Figs. 3, 4), leaving only the phonon blocking effect of localization. We plan to look at this possibility more carefully in future work.

Comment 8 of Reviewer 2: - *P-C. Wei et al. [Nature 539, E1–E2] points out that extensive and precise characterization of the related SnSe crystals are prerequisite for reproducibility and for drawing correct conclusions. Please give detailed information on sample preparation, characterization in the SI (e.g. density, in comparison to the theoretical one; structure refinement for all temperatures – and compare it to the structure obtained from the simulation; finally, as the most important for thermoelectric materials, please provide zT for all relevant temperatures)*

Reply: A structural analysis is now included in the revised paper in new Fig. 5. The derived lattice parameter at 300 K of 6.124 ± 0.0003 Å (the Al powder rings were used to calibrate the instrument assuming an aluminum lattice parameter of 4.046 Å) matches well the published value. In the Landolt-Bornstein reference book they report 6.124 Å as the value at 299 K, see https://link.springer.com/chapter/10.1007/10681727_903. The simulations are based on the published lattice parameter, but results are found to vary somewhat with small changes in the lattice parameter (see Ref. [10]).

In preparing the crystals we used Pb and Se materials from Alfa Aesar with 99.999% metals-based purity (a statement about starting material properties has been added to the Methods section on crystal growth). Care was taken to wash the silica ampoule with acids prior to the reaction to minimize the introduction of extrinsic impurities.

The Hall effect measurements described in the methods section provide a direct measure of charged impurities. In this case, Se vacancies are the most likely source. A concentration of Se vacancies (two charges each) of 0.004% would account for a carrier density of approximately $1.3 \times 10^{18} \text{ cm}^{-3}$, the order of magnitude observed here. Optimal doping for thermoelectric performance would require roughly two orders of magnitude more defects in a PbSe crystal, and

thus we believe that our PbSe crystal can be considered to be of high-quality in terms of how physical properties are impacted by defects, including phonon scattering. The intent was to isolate anharmonic effects, not optimize thermoelectric performance.

We performed measurements of the zT for our crystal at all relevant temperatures and now include these results in the SI:

A section was also added to the Methods, “Thermoelectric properties measurements”, which includes descriptions of the added thermal diffusivity, electrical resistivity, and Seebeck coefficient measurements (together these are used to determine a figure of merit zT curve). To go from thermal diffusivity to thermal conductivity we used the heat capacity data published here,

Pashinkin, A. S., Mikhailova, M. S., Malkova, A. S. & Fedorov, V. A. Heat capacity and thermodynamic properties of lead selenide and lead telluride. *Inorganic Mater.* **45**, 1226-1229 (2009).

This is Ref. [55] in our revised manuscript. The figure of merit of about 0.45 is decent for an undoped crystal but is not “high performance”, and this was not the aim of this study. To isolate intrinsic localization, we specifically avoided impurities and other defects to avoid confusion with localized vibrations associated with defects.

Comment 9 of Reviewer 2: - Title: “*Intrinsic anharmonic localization in thermoelectric PbSe*” and in the Abstract: “*Our study shows how strong anharmonicity can fundamentally alter vibrational transport properties even in perfect crystals.*” Is the small ILM or the optical mode meant in the Title? I assume the later one. It is not shown that the localized optical mode is caused by anharmonicity. This is neither predicted by the calculations, nor argued sufficiently.

Reply: As discussed above and in our revised manuscript we believe the small ILM and the localization of the optic phonon are directly related. The localization of the optic phonon is a consequence of the interaction of the phonon with the arrangement of the ILMs (see Reply to Comment 1 of Reviewer 2 above). The title is meant to refer to all the localization that occurs. Since there are no structural changes in either the experiment or simulations the resulting changes in the lattice dynamics can be attributed to anharmonicity. That we are seeing more than one manifestation of anharmonic localization is now stated explicitly at the beginning of the introduction.

Comment 10 of Reviewer 2: - *Abstract: “zero group velocity develops for the full spectral weight of transverse optic (TO) phonon. Arrest of TO phonon propagation...” These statements are misleading, the localised mode is shown only for a small part of the Brillouin zone.*

Reply: This is a good point. Normally ILMs just steal a small part of a phonon’s spectral weight and we were just trying to express the fact that here all the weight of the phonon goes flat in a part of the zone. We can now see how this could be misleading. In our revised manuscript we changed this to “...*zero group velocity develops abruptly for a significant section of the transverse optic (TO) phonon, which we attribute to a transition in the anharmonic dynamics.*” The flattening is observed about half way to the zone boundary at 0.5 r.l.u., which we think qualifies as a significant section. In the discussion we argue that the transition flattens the TO phonon through an interaction between the plane wave phonon and the ILMs where the phonons become trapped between ILMs, as described in the revised paper and in response to Comment 1 of Reviewer 2 above. However, this transition can also be thought of as a change in an interference pattern in extended mode instabilities, as described by Burlakov *PRL* **80**, 3988 (1998). We think the equivalence of these two different ways of describing the same physical phenomena is important enough that we also now mention it in the introduction.

Comment 11 of Reviewer 2: - *Abstract: “coincides with unusual sharpening of the longitudinal acoustic mode due to a loss of phase space for phonon scattering.” Sta*

Reply: It appears that this comment was cut short in our copy of this report.

Comment 12 of Reviewer 2: - *p.4.: “Our results reveal that localization occurs at temperatures close to predicted” It can be, but not shown. Ref. 8. does not state a clear temperature(range) where ILM gets separated from TO. It is also not shown, how sudden is that change.*

Reply: We agree that the temperature dependence is difficult to pin down and this is partly intrinsic to the behavior. Within the dynamical structure there are precursors. For example, there is the hint of a feature below the TO phonon in the data and simulations even at room temperature. With increasing temperature this evolves towards a flat mode. However, in the data there are also some more sudden changes, such as the flattening of the TO phonon. In our revised manuscript we show that this change can be traced to a subtle kind in the thermal diffusivity data at near 750 K. Unfortunately, it is not possible to measure phonon dispersion structures quickly enough to map the evolution of the pattern with such finely spaced temperatures.

With the ab initio simulation, it is also challenging to run a high density of temperatures. However, to determine if a similar transition to a flat TO more fragmented phonon eventually occurs at a higher temperature, we now include a calculation at 1000 K in a revised Fig. 2. Indeed our 1000 K simulation does show that the TO band further splits and flattens in going from 760 K to 1000 K,

Note that there is also a shift of TO phonon spectral weight away from the (113) zone center. This is interesting because a shift of TO phonon spectral weight away from the zone center is also evident in the triple axis measurements in going from 643 K to 793 K, see Fig. 1b,c. To clarify this we now also include these sentences in this section:

“Since uncertainties the simulation temperature scale exist¹⁰, we also include an additional simulation at 1000 K, Fig. 2e. This spectrum shows a flatter more fragmented TO phonon, more like the observations. Furthermore, there is a shift of TO phonon spectral intensity away from the (113) zone center, similar to a shift observed going from 643 K to 793 K in the triple axis measurement (Fig. 1b,c).”

Based on this, it seems likely that the temperature scales differ somewhat and therefore the measurements and calculations differ in part because of this. To clarify the uncertainty that this adds we also include the following sentences in the discussion section, *“Some challenges remain in fully reproducing such patterns from our ab initio simulations and there appears to be an offset in the temperature between theory and experiment.”*

Comment 13 of Reviewer 2: *“- Fig. 1.b: Can the uncommented branch at about 3 meV be ILM?”*

Reply: Without more data we cannot say for sure if that is a new feature or simply intensity from the TA phonon spreading up from below. However, the possibility that intensity from the TA phonon may shift to higher energies with the transition in the dynamical structure is actually supported by our simulations. At 1000 K the simulation in Fig. 2e predicts that the TA shifts to higher energies (stiffens) as the TO phonon flattens and further fragments. The physical reason for this is likely a relaxation of mode repulsion between the TO and TA phonons as the TO phonon moves up in energy. In our revised manuscript we now mention this feature in the text and note our uncertainty about its origin. First, on page 6 we state, *“There is also additional intensity appearing at the bottom of the spectrum near 3 to 4 meV, which may be a new feature or more likely intensity from the TA phonon spreading up from lower energies.”* Then

further down after the simulation results are presented, we add, “*There is also a stiffening of the TA phonon at 1000 K in the simulation, which supports the idea that the additional intensity appearing at the lowest energies in Fig. 1b is from the TA phonon shifting up into the measured range.*”

Comment 14 of Reviewer 2: -“*As shown in Fig. 1e,f a large anomalous decrease in the energy linewidths for the LA phonon measured along $Q = [H, H, 0]$ with heating from 643 K to 793 K” give fit details (parameters and restrictions) together with resolved fit spectra in SI, also for Fig.3.*”

Reply: Peak fitting procedure details for both a and IXS and INS scan are now included in SI. Both the peak positions and linewidths are in very good agreement between the IXS and INS measurements.

Comment 15 of Reviewer 2: -*Error bars should be either (visible) plotted or described in the figure caption.*

Reply: The error bars are now visible in the revised manuscript figures. We also added a comment in the caption, “*Error bars are statistical and represent one s.d.*”

Comment 16 of Reviewer 2: -“*relaxed out-of-plane angular (or Q) resolution of the BT7” what is the effect on the other branches? Can it be excluded that the ILM is an artefact? Ideally, a second picture of the calculated dispersion curves should be presented additionally (one of them in the SI), which includes all (Q and E) resolution effects.*”

Reply: The relaxed out-of-plane Q resolution had no effect on the peak positions in this case because all of the modes are flat in the out-of-plane $[1,-1,0]$ direction within the range of the Q resolution. The phonon energies measured using IXS with tight Q resolution and measured using INS with relaxed Q resolution agree precisely (see SI Figure 2), confirming that there is no effect on energies, just the intensity of the TA phonon (the transverse polarization component is decreased by about a factor of four in the IXS with the aperture fully open at about ± 0.5 degrees for HERIX compared to about ± 2 degrees for BT7). The flatness is even more clear for the TO and ILM features since $[1,-1,0]$ is nearly equivalent to the scanned $[110]$ direction near the (113) zone center and the modes are clearly flat in this direction. We looked at Q resolution during our experiment and concluded that energy resolution was a more important factor in the localization, which is why we turned to a cold neutron instrument.

Artefacts associated with the instrument are not normally temperature dependent and the ILM feature only appears at high temperatures. Also, a similar splitting in this TO mode was observed in PbTe (Ref. [8]), except without the flattening (localization) of the mode in Q . The simulations capture this difference, which would be surprising if it were an artefact. We also checked to see if the feature appears for negative neutron energy transfers on HYSPEC, and it does:

In this energy range the resolution is a little lower but the splitting/ILM feature is evident, even if less sharply defined. This image has now been added to SI, and it is mentioned in the results section on Fig. 2 as support for the existence of this feature. Artefacts do not appear the same for positive and negative neutron energy transfers, so this is a good test.

Comment 17 of Reviewer 2: - *A weak point of the work is the interaction with other researchers. Important references are missing, like:
recent review on PbSe: C. Gayner et al.: Materials Today Energy 9 (2018) 359-376
review about strategies for thermoelectric materials: M.Beekman et al.: Nature Materials 14 (2015) 1182-1185
doping effect on zT in PbSe: H. Wang et al.: PNAS 109/25 (2012) 9705-9709*

Reply: In our revised manuscript we now include these suggested references and many others increasing our overall reference count by 26, from 30 to 56.

Comment 18 of Reviewer 2: - *Please provide Grüneisen parameter(s), if possible both experimental and calculated ones.*

Reply: Grüneisen parameter(s), which characterize the quasiharmonic shift of a phonon frequency with volume, are not meaningful in the present context because the anharmonic effects are so far beyond the quasiharmonic limit that the phonons interact enough to interfere and generate entirely new features with increasing temperature. We could, in principle, apply a different set of parameters for different features appearing in different temperature ranges, but this does not seem to comply with the intent of the parameter, which is to provide a simplification.

Comment 19 of Reviewer 2: - *Methods/TAS INS: Please give details of data collection plan and comment it.*

Reply: To measure the transverse optic mode near the zone center, where an intrinsic localized mode (ILM) had been predicted, measurements were performed along $\mathbf{Q} = [H, H, 3]$ at eleven equally spaced points in H from the (113) zone center to $H = 0.75$, which is half way to the zone boundary. The longitudinal acoustic mode was measured along $\mathbf{Q} = [H, H, 0]$ at fifteen equally spaced points in H from the (220) zone center to $H = 2.35$. Scans were repeated at temperatures, $T = 300, 643, \text{ and } 793 \text{ K}$.

Comment 20 of Reviewer 2: - *Methods/ToF INS: Please give details to data reduction, e.g. software used, major steps done.*

Reply: To clarify how the data is combined to form the scattering function we added an extra phrase and reference to the HORACE software package: *A volume of data in Q - E space was obtained by rotating the angle between the [100] axis and the incident beam in 0.5° steps **and combining the data using the HORACE software package**³⁵. The data were collected at each angle from -45° to $+90^\circ$ to obtain a complete data set. Additional counting was done in the range from -90° to -20° to obtain better statistics in the region of particular interest within the (113) zone.*

Reviewer #3

Comment 1 of Reviewer 3: *This work reported experimental observation of phonon localization at elevated temperature using neutron scattering technique. The observed localization applied to the full spectrum of transverse optic mode, which is different than the theoretical prediction. In addition, sharpening of the longitudinal acoustic mode was discovered. These observations provide deeper understanding of the phonon transport in an important thermoelectric material, PbSe.*

Reply: We thank the reviewer for this overall positive assessment.

Comment 2 of Reviewer 3: *This work is worth publishing should the following comments be addressed properly*

1) The authors should report result on impurity measurement, and discuss potential effect on phonon scattering;

Reply: The purpose of these experiments was to isolate the contributions of anharmonic phonon interactions, so care was taken to minimize contributions from impurities. In preparing the crystals we used Pb and Se materials from Alfa Aesar with 99.999% metals-based purity (a statement about starting material purity has been added to the Methods section on crystal growth). Care was taken to wash the silica ampoule with acids prior to the reaction to minimize the introduction of extrinsic impurities.

The Hall effect measurements described in the methods section provide a direct measure of charged impurities. In this case Se vacancies are the most likely source. A concentration of Se vacancies (two charges each) of 0.004% would account for a carrier density of approximately $1.3 \times 10^{18} \text{ cm}^{-3}$, which is the order of magnitude observed here. Optimal doping for thermoelectric performance would require roughly two orders of magnitude more defects in a PbSe crystal, and thus we believe that our PbSe crystal can be considered to be of high-quality in terms of how physical properties are impacted by defects, including phonon scattering. This is further supported by new measurements of the thermoelectric properties, which indicate a maximum zT of ~ 0.45 . In our revised manuscript this is deduced from new measurements of the thermal diffusivity, electrical resistivity, and Seebeck coefficient of our crystal, see SI.

Comment 3 of Reviewer 3: and 2) *The authors should provide data on thermal conductivity or diffusivity.*

Reply: Thermal diffusivity measurements are now included in our revised manuscript. They are displayed in new Fig. 5 and are discussed in a new section entitled “Thermal diffusivity and structure analysis”. As discussed more fully in our reply to Comment 1 of Reviewer 2 these data allow us to more precisely pin down the temperature where the anharmonic dynamical structure transitions to a flat TO phonon.

We also added a new Methods section “Thermoelectric properties measurements”, which includes descriptions of new thermal diffusivity, electrical resistivity, and Seebeck coefficient measurements. Together these are used to determine a figure of merit zT curve for our crystal, which can be found in the SI.

Minor issues found in the manuscript

Comment 4 of Reviewer 3: 3) *Figure 1a should include visible error bars*

Reply: The error bars are now visible in the revised manuscript. We also added a comment in the caption, “***Error bars are statistical and represent one s.d.***”

Comment 5 of Reviewer 3: 4) *In line #7 of paragraph 2 on page 6, the citation of the figure should be "Fig. 2a".*

Reply: We have corrected this error.

REVIEWERS' COMMENTS:

Reviewer #2 (Remarks to the Author):

The authors put large efforts and succeed to significantly improve their manuscript: i) they added new experimental results to support their statements ii) they elaborated more deeply the high temperature range, with a special care of a possible phase transition iii) they could show that their simulation can reproduce the observed flattening of the TO mode and iv) extended the discussion significantly, in particular to explain the origin of this new feature and estimate its potential impact for future developments. Even if in case of the present crystal there is no significant effect on thermoelectric properties, the authors could argue convincingly, why this feature can be effective in other cases. All comments are satisfactorily replied. In summary, the authors present here a fundamentally new phenomena, which is ultimately connected to the thermoelectric performance. Therefore, I can now recommend the publication in Nature Communications.

There are only two minor remarks, which I would like to add:

- "temperature offset" - For me an offset in Temperature would mean that experimental data at T_{exp} are comparable with simulations at $T_{sim}=T_{exp}+const$. Is it really an offset, or might it be scaling or some unknown relation?

- "Recent ab-initio molecular dynamics calculations of PbSe that explicitly account for strong anharmonicity produce what appears to be an ILM forming at high temperatures and in resonance with the acoustic modes, but such a feature has not yet been reported experimentally." This is slightly misleading, since ILM was observed in NaI before, see ref. 28.

PS: Comment 11 is unfortunately also cut in my version. I'm sorry for that. I can not recover this point.

Fanni Juranyi

Reviewer #3 (Remarks to the Author):

The authors have address the two major comments from the reviewer. Information about the impurity is now included. Additional experiments were performed to measure the thermal diffusivity, corresponding discussion is included in the revised manuscript. These revisions are satisfactory.

Reviewer #2

Comment 1 of Reviewer 2: *“The authors put large efforts and succeed to significantly improve their manuscript: i) they added new experimental results to support their statements ii) they elaborated more deeply the high temperature range, with a special care of a possible phase transition iii) they could show that their simulation can reproduce the observed flattening of the TO mode and iv) extended the discussion significantly, in particular to explain the origin of this new feature and estimate its potential impact for future developments. Even if in case of the present crystal there is no significant effect on thermoelectric properties, the authors could argue convincingly, why this feature can be effective in other cases. All comments are satisfactorily replied. In summary, the authors present here a fundamentally new phenomena, which is ultimately connected to the thermoelectric performance. Therefore, I can now recommend the publication in Nature Communications.”*

Reply: We thank the reviewer for many helpful comments, a thoughtful review, and ultimately recommending publication.

Comment 2 of Reviewer 2: *There are only two minor remarks, which I would like to add:*

- *“temperature offset” - For me an offset in Temperature would mean that experimental data at T_{exp} are comparable with simulations at $T_{sim}=T_{exp}+const$. Is it really an offset, or might it be scaling or some unknown relation?*

Reply: The referee makes a good point. We change the wording to simply “*temperature difference*” between theory and experiment in our final revisions.

Comment 3 of Reviewer 2: - *“Recent ab-initio molecular dynamics calculations of PbSe that explicitly account for strong anharmonicity produce what appears to be an ILM forming at high temperatures and in resonance with the acoustic modes, but such a feature has not yet been reported experimentally.” This is slightly misleading, since ILM was observed in NaI before, see ref. 28.”*

Reply: We have removed the last part from the sentence “..., but such a feature has not yet been reported experimentally.” What we refer to here is that it is an ILM in the spectrum, not in the gap as observed before. The point is sufficiently made in the previous paragraph that this is really not needed, however.

Comment 4 of Reviewer 2: *“PS: Comment 11 is unfortunately also cut in my version. I’m sorry for that. I can not recover this point.”*

Fanni Juranyi”

Reply: No problem.

Reviewer #3

Comment of Reviewer 3: *“The authors have address the two major comments from the reviewer. Information about the impurity is now included. Additional experiments were performed to measure the thermal diffusivity, corresponding discussion is included in the revised manuscript. These revisions are satisfactory.”*

Reply: We thank the reviewer for helpful comments, suggestions, and a thoughtful review.